# Clinical process quality of antenatal care in low and middle-income countries: Cross-sectional evidence from 13 countries

**Roxanne Kovacs** *

Department of Economics, University of Gothenburg, Gothenburg, Sweden

* roxanne.kovacs@economics.gu.se

## Abstract

Increasing access to high-quality antenatal care is essential for reducing maternal and neonatal mortality in low- and middle-income countries (LMICs), yet systematic evidence on the quality of care remains limited. This study provides the broadest overview to date of the clinical process quality of antenatal care in LMICs, using nationally representative data from 13 countries (Afghanistan, Democratic Republic of Congo, Egypt, Ghana, Haiti, Kenya, Malawi, Namibia, Nepal, Rwanda, Senegal, Tanzania, and Uganda) to document levels and variability in quality and to examine differences between public and private health facilities. We analyse cross-sectional data from 21,850 antenatal consultations directly observed across these settings. Clinical quality is measured as the proportion of essential actions completed by healthcare providers during consultations, including history taking, physical exam-inations, and the recommendation of appropriate tests and treatments. Overall, the clinical quality of antenatal care is alarmingly low, with providers completing on aver-age only 39% of recommended actions (median 37.5, interquartile range 21.7–55.6). Quality varies substantially both within and between countries and is poorly predicted by commonly used proxies such as facility infrastructure, medication availability, or staffing levels. Although private for-profit facilities are generally better equipped and staffed and charge fees nearly five times higher than those in public facilities, there is no evidence that they provide higher-quality antenatal care. These findings indicate that the clinical quality of antenatal care remains low in resource-poor settings. It remains unclear to what extent increases in antenatal care utilization – long empha-sised in international development targets – will translate into improved maternal and neonatal outcomes, given the low quality of care currently being delivered. The results also provide no evidence to suggest that greater reliance on private-sector provision would improve quality.

**Data availability statement:** All data used in this study are from the Demographic and Health Surveys (DHS) Service Provision Assessment (SPA). These data can be requested directly from the DHS Program at https://dhsprogram.com/data/ The author is not permitted to share the data directly.

**Funding:** The authors received no specific funding for this work.

**Competing interests:** The authors have declared that no competing interests exist.

## Introduction

Rates of maternal mortality are unacceptably high in low and middle-income countries (LMICs). In 2020, close to 300,000 women died during pregnancy and childbirth and according to the World Health Organization, the majority of these deaths could have been prevented [1]. The largest share of maternal deaths globally (70%) occurred in Sub-Saharan Africa [1]. Reducing maternal and neonatal mortality has been at the forefront of many international development efforts, including the Sustainable Development Goals. There is a consensus amongst international policymakers that increasing uptake of maternity care – i.e., facility-based delivery, antenatal and postnatal care – is paramount for reducing deaths [2,3]. However, a growing body of evidence shows that increased utilisation of maternal health services does not, by itself, improve maternal and neonatal outcomes when the quality of antenatal and intrapartum care is poor. Low-quality antenatal care limits effective screening and management of conditions such as hypertensive disorders, anaemia, and infections, and reduces opportunities for timely referral and evidence-based preventive interventions, all of which are key pathways linking care quality to maternal and neonatal morbidity and mortality [4].

This study provides the broadest overview to date of the clinical process quality of antenatal care in LMICs and presents data on the actions taken by healthcare providers in 21,850 consultations in 13 LMICs (Afghanistan, Democratic Republic of Congo – DRC, Egypt, Ghana, Haiti, Kenya, Malawi, Namibia, Nepal, Rwanda, Senegal, Tanzania and Uganda), settings characterised by substantial workforce, resource, and governance constraints that may affect care quality [4–7].

Using a large multi-country dataset, this study also speaks to one of the key policy issues in healthcare provision in the world's poorest countries: whether care is best provided by the public or the private sector. Given the central role of antenatal care in identifying and managing pregnancy-related risks that contribute to maternal and neonatal mortality, differences in provider incentives across sectors may have important implications for the quality of care delivered during routine ANC consultations. The most common approach to delivering primary care in low-income settings (and in most of the world) has been via publicly-run facilities staffed by providers that are paid a fixed salary. One key argument against this approach is that healthcare workers have few incentives to invest effort during medical consultations – as they are paid the same regardless of the volume or quality of services provided – and that provider accountability for malpractice is generally low in LMICs. Hence, one might assume that the incentives of private-sector providers are better aligned with offering high-quality health care. Nonetheless, there is very little evidence on whether private-sector care is indeed superior to public provision in resource- poor settings [5,8–12]. This study is the first to compare the quality of antenatal care in public and private facilities in such a large number of countries.

The main outcome in the analysis is the proportion of relevant actions taken by healthcare providers during antenatal care consultations (history questions asked, physical examinations done, and tests and treatments recommended). The study

first describes the clinical process quality of antenatal care being provided in 13 LMICs and explores variation within and between countries. It then examines whether quality differs between public and private-for-profit facilities. Finally, it explores whether commonly used proxies of clinical quality (infrastructural quality, availability of equipment or staffing) are good predictors and investigates heterogeneity across settings.

## Methods

### Data

This multi-county cross-sectional study uses data from all available Service Provision Assessment (SPA) surveys that contain a module on the quality of antenatal care (see S1 Text). The SPA surveys collect nationally representative data in health facilities. The clinical quality of antenatal care is measured via direct clinical observations of medical consultations as well as patient exit interviews. This study includes data from all countries for which information on the quality of antenatal care is available: Afghanistan, DRC, Egypt, Ghana, Haiti, Kenya, Malawi, Namibia, Nepal, Rwanda, Senegal, Tanzania and Uganda. In five countries (DRC, Kenya, Nepal, Haiti and Uganda) a nationally representative sample of public and private-sector healthcare facilities was drawn.

### Outcome

The clinical process quality of antenatal care – the main outcome – is measured via direct clinical observations, where trained medical observers attend medical consultations with real patients and note down whether essential actions are performed by providers, using a standardised checklist. Clinical observations are arguably the best available method for measuring the clinical quality of pregnancy care. This is because data from clinical records are generally not detailed enough (and usually not available in LMICs) and because methods such as standardised patients (healthy people, posing as patients, who visit providers undercover) – the gold standard for measuring quality – are not suitable for this setting [13, 14]. In addition, there are clear guidelines on the actions providers need to take in all antenatal consultations, which can be captured in an observation checklist. The main concern with using data from clinical observations is a potential observation bias, or Hawthorne effect, whereby providers improve the quality of care they offer in response to being observed, meaning that data likely present an upper-bound estimate of the quality of antenatal care in LMICs [8]. Notably, in this setting there is no evidence to suggest that quality varies based on the order of consultations – which is how a Hawthorne effect is usually identified empirically (see S1 Text).

There is a core set of actions that healthcare providers must take during antenatal consultations. Whilst the checklists used during observations of antenatal consultations differed between SPA surveys, the measure of clinical quality used in this study captures a core set of items that are observed in nearly all countries and corresponds well with national and international guidelines for antenatal consultations – see S1 Text. Importantly, these actions represent the absolute minimum standard for antenatal care, as only the most essential actions are included.

The main outcome used in the analysis is the clinical process quality of care provided, defined as the proportion of essential history questions asked, physical examinations conducted, advice given, and preventive tests and treatments recommended or prescribed. Each action is weighted equally, reflecting their joint role as non-substitutable components of guideline-recommended antenatal care. The measure captures actions that should be completed at each antenatal visit, accounting for gestational age and whether the consultation is a first or follow-up visit.

### Facility and provider characteristics

SPA surveys in all countries include a questionnaire that is administered to the facility manager and to healthcare workers. The analysis controls for the following facility characteristics which are available for all countries: level of care (hospital, primary, below primary), infrastructural quality (availability of piped water and connection to the electric grid), external

supervision visits made to the facility, the availability of basic medical equipment (weighing scales, measuring tape, thermometer, stethoscope, blood pressure apparatus, light source, bag and mask and infusion kits), fees charged and the number of medical staff working at the facility. We distinguish between public and private-for-profit facilities. While this classification captures an important institutional distinction relevant for policy, it necessarily aggregates heterogeneous providers within each category, including differences in ownership structures, management practices, and contractual arrangements. As a result, estimates should be interpreted as average differences across broad sectors rather than as comparisons between uniform types of facilities.

The following provider characteristics are available for providers who conducted antenatal consultations: gender, cadre (doctor, nurse, midwife, medical officer, unskilled), work experience in the facility and relevant on-the-job training (related to antenatal care, child health, delivery or new-born care).

## Structural quality and human resources

This section shows healthcare provider and facility characteristics in 13 LMICs and provide an informative overview of the setting in which maternity care is being delivered in resource- poor settings. S1 Text shows information for each study country.

The basic infrastructure and equipment needed to deliver maternity care are lacking in most resource-poor settings – as only 48% (SD 0.50) of study facilities had piped water, 53% (SD 0.50) were connected to the electric grid and facilities had only 66% (SD 0.24) of the essential items of equipment to deliver antenatal care available. Despite the widely held view that workload is high in health facilities in LMICs, study data do not suggest that this is the case [15]. Based on a simple calculation detailed in S1 Text, where the total number of visits is compared to the number of skilled staff, this makes 10 consultations per skilled provider per day on average. Assuming that providers spend 15 minutes per consultation this would mean that providers spend on average approximately 2.5 hours a day consulting patients. Even though providers have other responsibilities, this suggests that there is considerable slack capacity. As shown in Table 1,

**Table 1. Facility and healthcare worker characteristics.**

| Facilities (N = 4,912): | Mean | SD | Providers (N = 6,155): | Mean | SD |
|---|---|---|---|---|---|
| Public | 0.69 | 0.46 | Female | 0.73 | 0.44 |
| Private for profit | 0.12 | 0.33 | Skilled provider | 0.89 | 0.32 |
| Non-profit | 0.16 | 0.37 | Doctor | 0.16 | 0.36 |
| Charges any fees | 0.83 | 0.38 | Medical officer | 0.04 | 0.20 |
| Below primary care | 0.04 | 0.18 | Nurse | 0.57 | 0.50 |
| Primary care facility | 0.67 | 0.47 | Midwife | 0.29 | 0.45 |
| Hospital | 0.28 | 0.45 | Work experience in facility | 7.24 | 8.45 |
| Has piped water | 0.48 | 0.50 | Years since graduation | 12.17 | 10.82 |
| Connected to electric grid | 0.53 | 0.50 | Any ANC training | 0.43 | 0.50 |
| Any external supervision | 0.97 | 0.17 | Any child-health training | 0.37 | 0.48 |
| % Basic equipment | 0.66 | 0.24 | Any delivery training | 0.39 | 0.49 |
| Nr. skilled staff | 15.01 | 134.36 | Any newborn training | 0.35 | 0.48 |
| Doctors | 0.12 | 0.21 | Nr.relevant training courses | 1.49 | 1.41 |
| Nurses | 0.65 | 0.31 | Any relevant training | 0.65 | 0.48 |
| Midwives | 0.21 | 0.26 | Self-reported weekly hours | 48.35 | 16.66 |
| Monthly visits (outpatient) | 1055.78 | 2404.67 | | | |
| Visits per skilled staff per day | 9.57 | 20.02 | | | |

Note: The table shows basic descriptive statistics for all healthcare facilities and providers in the sample. Data are based on SPA surveys from Afghanistan, DRC, Egypt, Ghana, Haiti, Kenya, Malawi, Namibia, Nepal, Rwanda, Senegal, Tanzania and Uganda.

antenatal care in LMICs is primarily provided by female (73%) skilled providers (nurses and midwives) with seven years of experience on average – although there might be some observation bias whereby skilled providers were favoured on the observation day. Even though provider workload, captured in terms of the number of consultations per provider per day is low, providers indicate that they work long hours (approximately 48 hours a week on average).

## Statistical analysis

Data from all antenatal consultations observed in the study countries were included – see Table A in S1 Text for sample sizes. The analysis was done in STATA 17 and proceeds as follows. First, the clinical process quality of care in the sample is described by plotting the, minimum and maximum, as well as the 25th and 75th percentile for the proportion of relevant actions done by healthcare providers for each of the study countries.

Second, differences in quality in public and private-for-profit facilities are examined in OLS regressions. Models take the form:

$$Y_{ifpc} = \alpha F_f + \beta P_p + \gamma PV_f + \delta_P PV_f \times \eta_c + \eta_c + \in_{ifpt}$$

where $Y_{ifpc}$ is the quality of healthcare received by patient $i$ in facility $f$ by provider $p$ in country $c$. Models include facility characteristics $F_f$ (public facility, primary care, access to water, access to electricity, supervision, availability of basic equipment) and provider and visit characteristics $P_p$ (gender, cadre, time of day, first visit), as well as country fixed effects $\eta c$. $PV_f$ captures whether a facility is private for-profit, rather than public or a not-for-profit and $PV_f \times \eta_c$ is the interaction between sector and the survey country. Marginal effects from the interaction term $PV_f \times \eta_c$ are presented. A pooled analysis, where all countries are included in the same model, is also implemented.

Third, the degree of correlation between clinical quality and facility and provider characteristics ($F_f$ and $P_p$) is explored in OLS regressions which are implemented for the whole sample (including country fixed effects $\eta c$) as well as separately for each country.

## Role of the funding source

The funder of the study had no role in study design, data collection, data analysis, data interpretation, or writing of the report.

## Results

### The quality of antenatal care in lmics

This section first provides an overview of the clinical process quality of antenatal care in resource-poor settings. It includes data from 21,850 consultations conducted in 4,912 healthcare facilities, representative of 13 LMICs (Afghanistan, DRC, Egypt, Ghana, Haiti, Kenya, Malawi, Namibia, Nepal, Rwanda, Senegal, Tanzania and Uganda). In this study, clinical quality is measured in terms of the proportion of essential actions taken by healthcare providers during antenatal consultations, which were observed by a medical professional, and represents an absolute minimum standard for antenatal care. Fig 1 shows the median, minimum and maximum, as well as the 25th and 75th percentile (lowest and highest quartile) for clinical quality.

Results suggest that the quality of antenatal care is critically low in resource-poor settings. Amongst the included countries, the median quality of antenatal care is lowest in Nepal and Senegal, where providers complete on average less than one out of five essential actions during antenatal consultations (Nepal: median 0.19, IQR: 0.12-0.28; Senegal: median 0.27, IQR: 0.17-0.39). Quality of care is highest in Rwanda, Ghana and Namibia – where providers still complete only between 55% and 66% of the most basic and essential actions during antenatal consultations (Rwanda: median,

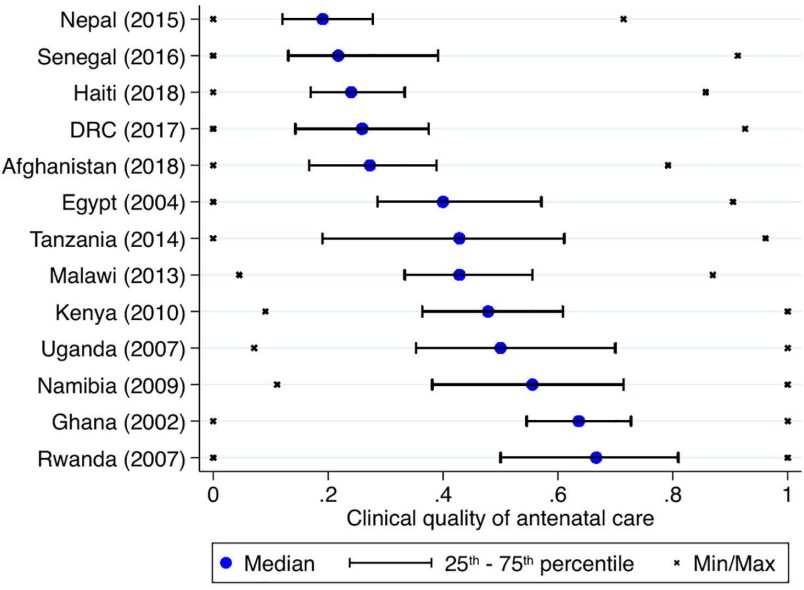

**Fig 1. Clinical process quality of antenatal care in 13 resource-poor settings.** Note: The figure shows country-level median quality of antenatal care (blue markers). The lines indicate the 25th and 75th percentile (highest and lowest quintile) and the crosses indicate the minimum/maximum for each country.

0.67 IQR: 0.50-0.81; Ghana: median, 0.63 IQR: 0.55-0.73; Namibia: median, 0.56 IQR: 0.38-0.71). Overall, whilst there is some overlap between higher economic development and higher clinical quality, there are important outliers, such as Rwanda (one of the poorest but best performing) and Egypt (the richest in the sample with quality only slightly above average). Fig A in S1 Text shows the quality of antenatal care at the consultation level for the whole sample. Across countries, providers complete on average only 39% (median 37.5, IQR 21.7-55.6) of recommended actions during antenatal consultations.

Besides the concerning low average levels of quality, the large variation in clinical quality – which can be well documented in this study given the large sample size and range of included settings – is striking. The interquartile range (25th – 75th percentile) generally overlaps even in the highest and lowest-performing countries – for instance, the best-performing providers in Senegal perform as well as the worst-performing providers in Namibia. In addition, the highest and lowest levels of quality observed lie close to 0% and 100% of checklist completion in all countries – suggesting potential equity issues in accessing high-quality care.

## Quality in the public and private sector

This section now focuses on differences in quality between sectors, using data from the five countries for which a representative sample of public and private-for-profit facilities were surveyed (DRC, Haiti, Kenya, Nepal and Uganda; covering 9,857 consultations). Any difference in quality between the two sectors reflects not only differences in incentives but also differences in provider, facility and patient characteristics. Nonetheless, this analysis provides an informative comparison of quality in the two sectors across a broad range of settings. As shown in Tables B and C in S1 Text, private-sector facilities are across the board, better equipped, both in terms of physical and human resources, to deliver high-quality antenatal care. In addition, women pay on average 5 international USD (SD 6.27) in public facilities, compared to 23 USD (SD 47.8) in the private for-profit sector – close to five times as much.

Fig 2 compares the clinical quality of antenatal care across sectors for each country, based on marginal effects from OLS regressions. There is no evidence to suggest that private-for-profit facilities provide higher quality care in any of the study countries. Coefficients are small in magnitude and not significant (coefficient estimates are 0.00 for Haiti, -0.03 for Kenya, 0.013 for Nepal and 0.06 for Uganda). Table D in S1 Text shows results from a pooled analysis, which shows similar results with point estimates close to zero and not significant. This result also holds when private non-profit organisations are removed from the sample (Table E in S1 Text). Hence, even though private for-profit facilities are on average better equipped and staffed and substantially more costly, results do not suggest that there is a benefit to patients in terms of higher clinical quality of care.

## Correlates of clinical quality

Table 2 examines the correlates of clinical process quality of antenatal care in LMICs. Results suggest that variation in the quality of care is difficult to predict and that individual correlates differ substantially across countries. As shown in column one, across the whole sample of countries, women and skilled providers deliver somewhat higher quality of care and quality is also somewhat higher in facilities with piped water and better access to basic equipment. However, the correlates of quality differ across settings, both in terms of significance and direction (also see S1 Text). For instance, in Afghanistan, clinical quality is *lower* in facilities with better structural quality (water, electricity or equipment). Moreover, it is notable that the largest share of the variation in clinical quality is unexplained by the included factors. This indicates that measures that are oftentimes used as proxies for clinical quality (availability of equipment, quality of infrastructure, presence of skilled providers) are generally not good predictors. There again appears to be considerable variation across settings, as the proportion of the variation in quality explained by the included factors is as high as 43% in Namibia (R-squared in column 9) and as low as 7% in Nepal (R-squared in column 10).

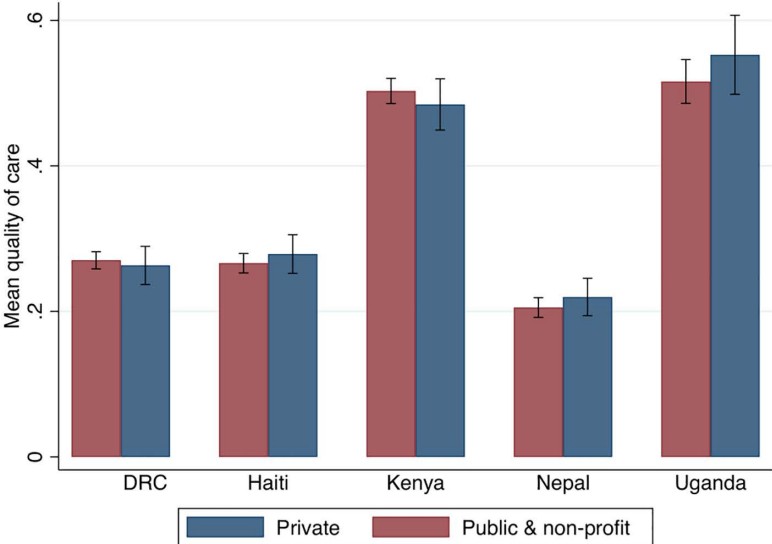

**Fig 2. Quality of healthcare in the public and private sector (marginal effects from OLS regressions).** Note: The bars show marginal effects from OLS regressions that include facility, provider and consultation-level controls (public facility, primary care, access to water, access to electricity, supervision, availability of basic equipment, gender, cadre, time of day, first visit). Models take the form: $Y_{ifpc} = \alpha F_f + \beta P_p + \gamma PV_f + \delta_P PV_f \times \eta_c + \eta_c + \in_{ifpt}$, where $Y_{ifpc}$ is the quality of healthcare received by patient *i* in facility *f* by provider *p* in country *c*. $PV_f$ captures whether a facility is private for-profit, rather than public or a not-for-profit and $PV_f \times \eta_c$ is the interaction between sector and the survey country. Marginal effects from the interaction term $PV_f \times \eta_c$ are presented.

**Table 2. Correlates of the clinical quality of antenatal care.**

| | (1) | (2) | (3) | (4) | (5) | (6) | (7) | (8) | (9) | (10) | (11) | (12) | (13) | (14) |
|---|---|---|---|---|---|---|---|---|---|---|---|---|---|---|
| | All | AF | CD | EG | GH | HT | KE | MW | NA | NP | RW | SN | TZ | UG |
| Primary care | -0.000 | -0.015 | 0.000 | 0.062** | 0.002 | -0.015 | 0.025 | -0.030* | -0.063 | -0.014 | 0.192*** | 0.052 | -0.015 | -0.024 |
| | (0.007) | (0.034) | (0.010) | (0.027) | (0.021) | (0.016) | (0.018) | (0.016) | (0.065) | (0.011) | (0.064) | (0.032) | (0.017) | (0.040) |
| Piped water | 0.023*** | -0.179*** | 0.015 | 0.117*** | -0.011 | 0.012 | 0.041** | 0.027 | -0.002 | 0.016* | 0.030 | -0.008 | 0.016 | 0.000 |
| | (0.008) | (0.035) | (0.011) | (0.032) | (0.018) | (0.016) | (0.017) | (0.022) | (0.051) | (0.009) | (0.031) | (0.031) | (0.016) | (0.040) |
| Electric grid | 0.011 | -0.078* | 0.022** | 0.061* | 0.001 | -0.018 | 0.007 | 0.005 | -0.009 | -0.003 | 0.049 | 0.046** | 0.011 | 0.009 |
| | (0.007) | (0.042) | (0.011) | (0.031) | (0.019) | (0.012) | (0.017) | (0.015) | (0.022) | (0.013) | (0.031) | (0.022) | (0.016) | (0.029) |
| Supervision | -0.011 | 0.006 | 0.101*** | 0.002 | -0.013 | -0.015 | -0.045 | 0.028 | -0.049 | -0.013 | 0.031 | 0.057** | 0.024 | . |
| | (0.015) | (0.039) | (0.025) | (0.051) | (0.053) | (0.018) | (0.058) | (0.038) | (0.050) | (0.021) | (0.085) | (0.022) | (0.052) | |
| Basic equipment | 0.008** | -0.042** | 0.016*** | 0.011 | 0.014 | 0.008 | 0.010 | 0.004 | 0.002 | 0.004 | 0.041* | 0.013 | 0.029*** | 0.010 |
| | (0.004) | (0.018) | (0.006) | (0.016) | (0.013) | (0.006) | (0.017) | (0.008) | (0.011) | (0.005) | (0.021) | (0.011) | (0.006) | (0.023) |
| Public | -0.012 | -0.115*** | -0.006 | 0.042 | 0.020 | 0.009 | -0.015 | -0.018 | 0.028 | -0.015 | 0.062* | -0.003 | 0.046*** | -0.030 |
| | (0.007) | (0.036) | (0.011) | (0.040) | (0.020) | (0.013) | (0.016) | (0.014) | (0.034) | (0.013) | (0.037) | (0.031) | (0.017) | (0.034) |
| Female | 0.018*** | . | 0.001 | -0.009 | -0.034 | 0.021** | 0.054*** | 0.015 | 0.033 | 0.007 | 0.039 | 0.128*** | 0.005 | 0.053 |
| | (0.006) | | (0.009) | (0.021) | (0.047) | (0.010) | (0.017) | (0.014) | (0.024) | (0.013) | (0.046) | (0.031) | (0.020) | (0.083) |
| Skilled | 0.041*** | . | 0.047** | 0.051 | 0.045** | 0.015 | 0.153*** | 0.013 | 0.077*** | -0.014 | 0.003 | -0.036 | 0.103*** | 0.071 |
| | (0.009) | | (0.018) | (0.054) | (0.022) | (0.016) | (0.035) | (0.027) | (0.030) | (0.020) | (0.038) | (0.032) | (0.016) | (0.047) |
| Observations | 21,388 | 494 | 4,517 | 1,062 | 1,525 | 1,528 | 1,437 | 2,083 | 825 | 1,565 | 732 | 849 | 4,010 | 761 |
| R-squared | 0.444 | 0.225 | 0.198 | 0.119 | 0.058 | 0.112 | 0.193 | 0.094 | 0.437 | 0.070 | 0.145 | 0.154 | 0.173 | 0.213 |

Note: The table shows correlates of the clinical quality of antenatal care. Column 1 includes all countries in the sample, including a simple sampling weight (equal to 1,000/observations, reflecting that roughly 1,000 consultations were observed in most countries) and country fixed-effects. The remaining columns show results for individual countries: Afghanistan (AF), Democratic Republic of the Congo (CD), Egypt (EG), Ghana (GH), Haiti (HT), Kenya (KE), Malawi (MW), Namibia (NA), Nepal (NP), Rwanda (RW), Senegal (SN), Tanzania (TZ), Uganda (UG). In Afghanistan (column 2) all providers were skilled and female and in Uganda (column 14) all facilities were externally supervised. Facilities sampled in some countries were below primary-care level, this category is included but not shown – meaning that hospitals are the comparison category for facility type. *** p < 0.01, ** p < 0.05, * p < 0.1. Robust standard errors, clustered by facility, in parentheses.

S1 Text examines whether the amount paid by women, which varies considerably between consultations, reflects differences in clinical quality. Table F in S1 Text shows the increase in clinical quality of care associated with a one standard deviation increase in the amount paid and, overall, there is no evidence for a significant link between the two.

Taken together, the results show that the clinical process quality of antenatal care in LMICs is low and highly variable within and across countries, and that this variation is poorly explained by commonly used structural proxies such as infrastructure, equipment, staffing, or prices. Importantly, there is no evidence to suggest that private for-profit facilities provide higher-quality antenatal care than public facilities, despite being better resourced and substantially more costly on average.

## Discussion

This study provides the broadest overview to date of the clinical process quality of antenatal care in resource-poor settings. Results suggest that the quality of antenatal care being provided in LMICs is low as providers complete on average only 39% (median 37.5, IQR 21.7-55.6) of essential actions during consultations and that quality varies widely both within and between countries. The finding that the process quality of antenatal care is low in LMICs is in line with previous work, usually conducted in only one country or setting [5,8–11] and an overview of seven SPA surveys [12] that does not match observation checklists across settings and therefore cannot compare providers in different countries based on the same set of actions – as is the case here. The findings from this study make an important contribution to the literature by

generalising and benchmarking those of previous studies. In addition, the results underscore that whilst poor quality antenatal care is common in LMICs, there is considerable variation amongst resource-poor countries. The finding that quality of care varies widely within countries also provides novel large-scale evidence on potential equity issues in accessing high-quality antenatal care within a given country.

Whilst there is an ongoing debate about whether primary care in LMICs is best delivered by salaried public-sector providers or in for-profit private facilities [16–18] only a small number of studies have provided empirical evidence on differences in quality between the two sectors. Existing evidence is mixed [12,19–21] and previous studies are limited in that they often do not distinguish between private for- profit facilities and private not-for-profit facilities – although incentives differ drastically. A study in India, where standardised patients were sent to the same doctors working in their public and private-sector practice, found that the same providers perform better in the private sector – suggesting that incentives are better aligned with providing high-quality care in the private secto [22]. No previous study has provided an overview of the clinical quality of care in the public and private sector in a representative sample of facilities across such a large number of countries. This study provides no evidence to suggest that private-sector providers provide higher quality of care in the broad range of countries studied. This is particularly striking as private-sector facilities are on average better equipped and staffed than their public-sector counterparts and women pay nearly five times as much during consultations.

This study is limited in a number of respects. First, estimates of the quality of antenatal care are subject to an observation bias. Hence, even though average levels of quality of care are very low, these estimates likely present upper bounds. In addition, the analysis cannot fully account for unobserved provider characteristics – such as intrinsic motivation, experience, or clinical knowledge – that may influence performance and contribute to residual variation in measured quality. Second, it only provides information on the quality of antenatal care and results may be specific to this type of service. However, given the repeat nature of antenatal consultations, providers should have additional incentives to invest adequate effort. Moreover, focusing on the quality of antenatal care is highly policy-relevant given the high levels of maternal and neonatal mortality in LMICS. Finally, we only focus on the clinical quality of care and disregard other less technical aspects of the quality of services – such as provider demeanour or bedside manner, as well as outcome indicators such as mortality. Process quality indicators also capture adherence to guideline-recommended actions but may not fully reflect appropriate clinical judgement or case-specific decision-making in all circumstances. Nonetheless, the clinical quality of care plays a key role in determining the degree to which uptake of antenatal care influences patient outcomes and is therefore paramount for reaching international development targets around maternal and neonatal health.

There are five key implications based on the study findings. First, the clinical process quality of healthcare is critically low in many LMICs. To reach key policy targets around women's health and maternal mortality, clinical process quality needs to improve. However, existing evidence provides limited and inconclusive guidance on how best to achieve sustained improvements in provider performance in routine care settings. While a wide range of intervention – such as in-service training, supervision, audit and feedback, and incentives – have been implemented, systematic reviews suggest that their effects on clinical practice are often modest, highly context-specific, and difficult to sustain at scale [23,24]. As a result, improving clinical process quality remains a major unresolved challenge in health systems in LMICs.

Second, the clinical quality of antenatal care is difficult to predict using frequently used indicators (such as infra-structural quality, staffing or the availability of drugs and equipment) and the correlates of quality vary substantially between settings. This suggests that to adequately capture clinical quality of care, it needs to be measured directly. More widespread measurement – such as that provided by facility-based surveys like the SPA – is therefore critical for understanding and tracking clinical quality over time. Recent disruptions to, and uncertainty about, the continuation of survey programmes like the DHS are a cause for concern, given the central role these data play in informing research and policy on the quality of care in LMICs.

Third, the clinical quality of antenatal care varies widely within countries. This points to potential equity issues in accessing high-quality care. Further research is needed on the degree to which patient or local-area characteristics account for these differences, and which policy responses are effective at creating equitable access to high-quality care.

Fourth the essential infrastructure and items of equipment needed to deliver antenatal care and other primary health-care services are oftentimes not readily available in healthcare facilities. This is disappointing given existing efforts to improve service readiness in LMICs and highlights a need for further research in this area.

Finally, based on the broad overview of the quality of care presented in this paper, private-sector provision does not appear to be a straightforward alternative for improving the quality of care in LMICs. Hence, policymakers should be careful with expanding private-sector provision in the hope that this will improve process quality of care.

## Key points

**Question:** What is the clinical process quality of antenatal care in low and middle-income countries (LMICs)?
**Findings:** Clinical quality of antenatal care in LMICs is alarmingly low, with healthcare providers completing only 39% of essential actions during consultations. There is significant variability in quality both within and between countries, and private for-profit facilities – despite being better equipped and staffed – do not provide higher quality healthcare.
**Meaning:** The low level of clinical quality raises concerns about the extent to which increased uptake of antenatal care—an explicit focus of international targets—will translate into improved maternal and neonatal outcomes.

## Supporting information

**S1 Text. Supplementary methods and results.** This document provides additional methodological detail and supplementary results. It includes information on the Service Provision Assessment (SPA) surveys used in the analysis; sample sizes and country coverage; construction of the clinical process quality measure and the checklist items included; assessment of consultation order effects; supplementary tables and figures; and additional analyses comparing public and private facilities.
(PDF)

## Author contributions

**Conceptualization:** Roxanne Kovacs.

**Data curation:** Roxanne Kovacs.

**Formal analysis:** Roxanne Kovacs.

**Funding acquisition:** Roxanne Kovacs.

**Investigation:** Roxanne Kovacs.

**Methodology:** Roxanne Kovacs.

**Writing – original draft:** Roxanne Kovacs.

**Writing – review & editing:** Roxanne Kovacs.

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
