## [Decision Letter · Decision Letter 0]

21 Jan 2026

PGPH-D-25-02697

Clinical process quality of antenatal care in low and middle-income countries: Cross-sectional evidence from 13 countries

Dear Dr. Kovacs,

Thank you for submitting your manuscript to PLOS Global Public Health. After careful consideration, we feel that it has merit but does not fully meet PLOS Global Public Health’s publication criteria as it currently stands. Therefore, we invite you to submit a revised version of the manuscript that addresses the points raised during the review process.

We look forward to receiving your revised manuscript.

Kind regards,

Shiyam Sunder, MBBS, MSc epidemiology, PhD

Academic Editor

Journal Requirements:

Additional Editor Comments (if provided):

Reviewers' comments:

Reviewer's Responses to Questions

**Comments to the Author**

1. Does this manuscript meet PLOS Global Public Health’s publication criteria?

Reviewer #1: Yes

Reviewer #2: Yes

2. Has the statistical analysis been performed appropriately and rigorously?

Reviewer #1: Yes

Reviewer #2: Yes

3. Have the authors made all data underlying the findings in their manuscript fully available (please refer to the Data Availability Statement at the start of the manuscript PDF file)?

Reviewer #1: Yes

Reviewer #2: Yes

4. Is the manuscript presented in an intelligible fashion and written in standard English?

Reviewer #1: Yes

Reviewer #2: Yes

Reviewer #1: This manuscript addresses an important and highly relevant public health question: the quality of clinical processes during antenatal care (ANC) in low- and middle-income countries (LMICs), and differences between public and private-for-profit providers. Using a large, multi-country dataset covering 21,850 ANC consultations across 13 LMICs, the study makes a valuable contribution to the literature by focusing on observed provider actions rather than service utilization alone. Overall, the manuscript is methodologically sound, the analyses are appropriate, and the conclusions are generally well supported by the data. I recommend minor revisions to improve clarity, structure, and interpretability.

Below are specific comments organized by manuscript section.

Abstract

The abstract is clear and informative; however, it could be strengthened by briefly clarifying how “clinical process quality” is operationalized (e.g., proportion of recommended history questions, examinations, tests, and treatments performed).

Consider adding a short phrase highlighting the policy relevance of comparing public versus private-for-profit facilities, particularly in resource-constrained settings.

Introduction

The introduction provides strong global context regarding maternal mortality and the importance of antenatal care. However, the flow could be improved for clarity and coherence.

Consider more explicitly defining the link between poor-quality antenatal care and maternal and neonatal outcomes, potentially supported by a small number of additional references.

The discussion of LMICs could be expanded slightly to clarify the structural and systemic challenges affecting care quality (e.g., workforce constraints, accountability mechanisms, resource limitations).

While the rationale for comparing public and private providers is well articulated, the introduction would benefit from a clearer transition that explicitly links maternal mortality, ANC quality, and provider incentives.

The number of references is acceptable; however, adding a few targeted citations (particularly on ANC quality and provider performance in LMICs) could strengthen the theoretical framing. This is a suggestion rather than a requirement.

Methods

The data sources, sampling strategy, and inclusion criteria are clearly described and appropriate for the research question.

The construction of the main outcome variable (proportion of relevant clinical actions performed) is reasonable and well motivated. A brief justification for weighting all actions equally (if applicable) could further strengthen this section.

The classification of facilities into public and private-for-profit is clear; however, a short discussion acknowledging heterogeneity within these categories would improve transparency.

The statistical approach is appropriate. Minor clarifications regarding model specification (e.g., choice of controls, fixed effects) would enhance reproducibility, but no major methodological concerns were identified.

Results

Results are presented clearly and logically, with appropriate use of tables and figures.

The description of cross-country variation in ANC quality is informative and highlights important heterogeneity.

When discussing differences between public and private-for-profit facilities, it may be helpful to more explicitly emphasize effect sizes and their practical significance, not only statistical significance.

Consider briefly restating the key findings at the end of the Results section to aid reader comprehension.

Discussion

The discussion appropriately interprets the findings and avoids overstatement.

The policy implications are relevant and well aligned with the study results; however, the discussion could be strengthened by more explicitly linking findings to potential health system interventions (e.g., provider incentives, supervision, quality monitoring).

The limitations section is appropriate, though it could be expanded slightly to emphasize issues such as unobserved provider characteristics or potential measurement limitations of process quality indicators.

The conclusion is supported by the data and appropriately cautious.

Presentation and Language

The manuscript is written in clear, standard English and is generally well organized.

Minor typographical and formatting issues are present and should be addressed during revision, but these do not detract from the scientific content.

Overall Assessment

This is a strong and policy-relevant manuscript that makes a meaningful contribution to the literature on maternal health and healthcare quality in LMICs. The suggested revisions are primarily focused on improving clarity, structure, and contextualization rather than addressing substantive methodological concerns. I therefore recommend acceptance pending minor revisions.

Reviewer #2: Great analysis. Nothing to add. Absolutely interesting topic especially for 13 countries included in the study. Antenatal care unfortunately, still is big problem for low and some of the middle income countries all over the world. You address this problem clearly though meticulously analysing multiple factors. So conclusions are clear.

**Do you want your identity to be public for this peer review?** For information about this choice, including consent withdrawal, please see our Privacy Policy

Reviewer #1: No

Reviewer #2: **Yes:** Igor Aluloski

---

## [Editor Report · Decision Letter 1]

29 Jan 2026

Clinical process quality of antenatal care in low and middle-income countries: Cross-sectional evidence from 13 countries

PGPH-D-25-02697R1

Dear Dr Kovacs,

We are pleased to inform you that your manuscript 'Clinical process quality of antenatal care in low and middle-income countries: Cross-sectional evidence from 13 countries' has been provisionally accepted for publication in PLOS Global Public Health.

Best regards,

Shiyam Sunder, MBBS, MSc epidemiology, PhD

Academic Editor